# High Levels of the Cleaved Form of Galectin-9 and Osteopontin in the Plasma Are Associated with Inflammatory Markers That Reflect the Severity of COVID-19 Pneumonia

**DOI:** 10.3390/ijms22094978

**Published:** 2021-05-07

**Authors:** Gaowa Bai, Daisuke Furushima, Toshiro Niki, Takashi Matsuba, Yosuke Maeda, Atsushi Takahashi, Toshio Hattori, Yugo Ashino

**Affiliations:** 1Research Institute of Health and Welfare, Kibi International University, Takahashi 716-8508, Japan; gaowabai@kiui.ac.jp (G.B.); atakah7@kiui.ac.jp (A.T.); 2Department of Drug Evaluation and Informatics, Graduate School of Pharmaceutical Sciences, University of Shizuoka, Shizuoka 422-8526, Japan; dfuru@u-shizuoka-ken.ac.jp; 3Department of Immunology, Kagawa University, Kagawa 761-0793, Japan; niki@med.kagawa-u.ac.jp; 4Department of Microbiology and Immunology, Faculty of Medicine, Tottori University, Tottori 683-8503, Japan; matsubat@phoenix.ac.jp; 5Department of Animal Pharmaceutical Science, School of Pharmaceutical Science, Kyusyu University of Health and Welfare, Nobeoka, Miyazaki 882-8508, Japan; 6Viral Section, Department of Microbiology, Faculty of Life Sciences, Kumamoto University, Kumamoto 860-8556, Japan; ymaeda@gpo.kumamoto-u.ac.jp; 7Department of Respiratory Medicine, Sendai City Hospital, Miyagi 982-8502, Japan

**Keywords:** COVID-19, COVID pneumonia, infectious diseases, full-length galectin-9, truncated galectin-9, full-length osteopontin, undefined osteopontin, tocilizumab, inflammatory markers, therapy

## Abstract

Numbers of patients with coronavirus disease 2019 (COVID-19) have increased rapidly worldwide. Plasma levels of full-length galectin-9 (FL-Gal9) and osteopontin (FL-OPN) as well as their truncated forms (Tr-Gal9, Ud-OPN, respectively), are representative inflammatory biomarkers. Here, we measured FL-Gal9, FL-OPN, Tr-Gal9, and Ud-OPN in 94 plasma samples obtained from 23 COVID-19-infected patients with mild clinical symptoms (CV), 25 COVID-19 patients associated with pneumonia (CP), and 14 patients with bacterial infection (ID). The four proteins were significantly elevated in the CP group when compared with healthy individuals. ROC analysis between the CV and CP groups showed that C-reactive protein had the highest ability to differentiate, followed by Tr-Gal9 and ferritin. Spearman’s correlation analysis showed that Tr-Gal9 and Ud-OPN but not FL-Gal9 and FL-OPN, had a significant association with laboratory markers for lung function, inflammation, coagulopathy, and kidney function in CP patients. CP patients treated with tocilizumab had reduced levels of FL-Gal9, Tr-Gal9, and Ud-OPN. It was suggested that OPN is cleaved by interleukin-6-dependent proteases. These findings suggest that the cleaved forms of OPN and galectin-9 can be used to monitor the severity of pathological inflammation and the therapeutic effects of tocilizumab in CP patients.

## 1. Introduction

Severe acute respiratory syndrome coronavirus 2 (SARS-CoV-2) [1] caused a pandemic of coronavirus disease 2019 (COVID-19) with more than 125 million cases and more than 2.75 million deaths as of the end of March 2021. Severity is highly variable, ranging from asymptomatic infections, mild cold symptoms, severe pneumonia to respiratory failure requiring mechanical ventilation and death from multiple organ failure [2]. Risk factors for aggravation have been clarified including older age, smoking, obesity, and pre-existing conditions such as hypertension, diabetes mellitus, cardiovascular diseases, chronic lung diseases, cancer, and chronic kidney disease [3]. However, even if the patients have mild symptoms at their initial visit to the clinic, they may suddenly develop fatal acute respiratory syndrome and/or multiple organ failure over the course of the illness [4]. Biomarkers are strongly desired that can predict the final severity of COVID-19 in the early stages of SARS-CoV-2 infection.

Acute respiratory syndrome is caused or accompanied by cytokine storm [5], where high levels of cytokines and proinflammatory molecules are present in the plasma. These molecules are thought to cause tissue injury, especially in the lungs [6]. The monitoring of cytokines including interleukin-6 (IL-6), IL-10, and tumor necrosis factor-α was recommended for the early detection of severe disease in patients [7]. Levels of IL-6 correlated with COVID-19 severity and IL-6 has a key role in cytokine storm and the inflammatory cascade [6,8]. Signaling inhibitors of IL-6 are candidate drugs for cytokine storm and tocilizumab (TCZ), a humanized monoclonal antibody that recognizes membrane-bound and soluble IL-6 receptors, which might be useful to treat COVID-19 pneumonia. A previous study of TCZ administration showed a significant clinical improvement in COVID-19 patients with pneumonia requiring a ventilator [9,10]. However, clinical improvement and mortality were not improved by TCZ therapy [11], and ICU admission and mortality rates were not reduced [12,13]. It should be noted that TCZ therapy is associated with severe infections [14], and a possible correlation between TCZ therapy and medication-related osteonecrosis of the jaws was indicated [15].

Detailed immunological analyses of COVID-19 patients showed significant increases in proinflammatory or anti-inflammatory cytokines, including T helper type-1 and type-2 cytokines, chemokines, and galectins. Galectin (Gal)-1, Gal-3, and Gal-9 were increased in patients compared with controls [16,17]. In addition, high plasma levels of granulocyte macrophage colony stimulating factor, IL-18, C-C motif chemokine 2, C-X-C motif chemokine ligand 10, and osteopontin (OPN) confirmed the importance of monocytes in pneumonia associated with COVID-19 [18]. Gal-9 and OPN are matricellular proteins that interact with cellular receptors and proteases [19,20]. The full-length Gal-9 (FL-Gal9) is the active form and the cleavage by proteases degrades the activity [21,22], while the cleaved form of OPN demonstrates distinct immunological properties compared with the FL-OPN [23]. We reported that FL-Gal9 was elevated in the plasma of patients with acute HIV [24], dengue [25], or malaria [26] and that their levels reflected disease severity. Furthermore, the FL and cleaved forms of OPN were elevated in the plasma of dengue patients [27]. Gal-9 is cleaved by neutrophil elastase, matrix metalloproteinase (MMP)-3 [28], and thrombin [29], and OPN is cleaved by thrombin, MMP-3, MMP-7, and MMP-9 [30,31]. Thrombin is involved in COVID-19-associated coagulopathy and is highly expressed in inflamed lesions and sites of tissue remodeling [32,33]. These enzymes might cleave Gal-9 and/or OPN in inflamed tissues; therefore, we measured their FL and cleaved forms to provide in-depth pathophysiological information on COVID-19 patients. We previously reported that analysis by enzyme-linked immunosorbent assay (ELISA) differentiated between the levels of the cleaved form of OPN (undefined (Ud)-OPN) and FL-OPN [27]. Recently, we established the ELISA system which can differentiate the truncated form of Gal-9 (Tr-Gal9) and FL-Gal9 [34], and reported that plasma levels of Tr-Gal9 reflected inflammation and the severity of disease in acquired immunodeficiency syndrome (AIDS) and AIDS associated with tuberculosis (AIDS/TB) patients [35]. 

As a systemic inflammatory marker, C-reactive protein (CRP) was associated with disease development and showed good performance in predicting severity in an early stage of COVID-19 [36]. CRP is known to be synthesized by IL-6-dependent and -independent pathways [37]. Cytokine, soluble interleukin-2 receptor α (sIL-2R), also known as CD25, released mainly from lymphocytes and monocytes, appears to play a role in the biology of COVID-19 and reflects its severity [38,39]. Patients with COVID-19 with markedly elevated d-dimer levels may require hospitalization, despite the severity of clinical presentation, according to the International Society of Thrombosis and Hemostasis guideline [40]. The elevations of d-dimer and ferritin, another inflammatory coagulation marker, were also known to be associated with poor outcome of the patients [41]. It is known that kidney diseases are associated with COVID-19 infection and creatinine levels are elevated in these patients [42]. Patients with elevated urinary β_2_-microglobulin (B2M) and creatinine levels showed lower rates of discharge [43].

In this study, we measured the levels of FL-OPN, FL-Gal9, and their truncated forms in COVID-19 patients and investigated the correlation with the above clinically commonly used indicators of inflammation, renal function, and abnormal coagulation. We also determined whether they reflect clinical severity and the therapeutic efficacy of TCZ in COVID-19 patients.

## 2. Results

### 2.1. Clinical Findings

Febrile patients were recruited from the Outpatient Department of Sendai City Hospital (SCH) from July 2020 to October 2020. COVID-19-infected patients were divided into patients with mild clinical symptoms (CV), COVID-19 patients associated with pneumonia (CP). Bacterial-infected patients not infected with COVID-19 (ID) were also studied (Figure 1).

There were 23 patients in the CV group, 25 in the CP group, and 14 in the ID group. There were significant differences in age, sex, aminotransferase, CRP, and serum albumin levels were significantly different between the three groups (Table 1).

CP patients suffered from complications including hypertension, hyperlipidemia, diabetes mellitus, and cerebral infarct. More patients in the CP group had clinical symptoms including cough, diarrhea, and dyspnea compared with patients in the CV and ID groups. The severity of symptoms in each patient was assessed with reference to the WHO classification [44] (Table 2).

### 2.2. Levels of Gal-9 and OPN in Patients

The levels of Gal-9 and OPN in the groups and the healthy control (HC) group were compared (Figure 2). The levels of Tr-Gal9, FL-OPN, and Ud-OPN in the CV group were significantly higher than in the healthy controls (HC) group. The levels of all four proteins in the CP group were significantly higher than in the HC group. Only FL-Gal9 and Tr-Gal9 had higher levels in the CP group compared with the CV group. The levels of Tr-Gal9, FL-OPN, and Ud-OPN were significantly higher in the ID group compared with the HC group. The levels of FL-Gal9 in the ID group were significantly lower than in the CP group, and the Ud-OPN levels in the ID group were significantly higher compared with the CV group. Ratios of Tr-Gal9/FL-Gal9 showed no significant changes between the HC, CV, and CP groups, but its ratio is significantly lower in the CP group as compared with the ID group (Appendix A). Ud-OPN/FL-OPN ratios showed the highest in the ID group, and the ratios of the ID and CP groups were significantly higher than that of the HC group. There were no significant differences between the CV and CP groups.

### 2.3. Levels of Inflammatory, Coagulation, Kidney and Respiratory Indicators in COVID-19 Patients

The levels of CRP, sIL-2R, and ferritin in the CP group were significantly increased compared with those in the CV group (Figure 3A,C,D). Similarly, the levels of percutaneous oxygen saturation (SpO_2_), the SpO_2_ fraction of inspiratory oxygen (FiO_2_) (S/F) ratio, and the numbers of lymphocytes were significantly lower in the CP group compared with the CV group (Figure 3B,H,I). The levels of CRP, sIL-2R, d-dimer, and B2M in the ID were significantly increased compared with the CV group (Figure 3A,C,E,F).

### 2.4. Receiver Operating Characteristic (ROC) Analysis of Inflammatory, Coagulation, Kidney and Respiratory Indicators

Area under curve (AUC) values were obtained by ROC analysis between CV, CP, ID and HC. Detailed analytical results are shown in Appendix A FL- and Ud-OPN had the highest AUC values (>0.97), followed by Tr-Gal9 (0.88) in the CV group. Ud-OPN and Tr-Gal9 had very high values (>0.99) in the CP group indicating cleavage occurred in this group. FL- and Ud-OPN values were 1.00 in the ID group, indicating the significant elevation of OPN during bacterial infection (Figure 4A–C).

It is important to detect the development of pneumonia; therefore, ROC analysis was performed between the CV and CP groups and we compared Gal-9 and OPN levels with inflammatory, coagulation, and kidney indicators commonly used in clinical practice (Figure 2). The AUC of Gal-9 and OPN showed that Tr-Gal9 had the highest value (0.89), followed by Ud-OPN (0.81), FL-Gal9 (0.80), and FL-OPN (0.70) (Figure 5A). The ROC curve of other inflammatory markers showed that CRP had the highest AUC value (0.94), followed by ferritin (0.88), and sIL-2R (0.76) (Figure 5B). The SpO_2_ and SpO_2_/FiO_2_ values were 0.76 and the creatinine and lymphocytes values were below 0.70 (Figure 5C).

### 2.5. Correlations between Inflammatory, Coagulation, Kidney and Respiratory Indicators

To understand the relevance of the elevated levels of Gal-9 and OPN in three groups, they were compared with inflammatory and respiratory markers. In the CV group, FL-Gal9 and FL-OPN were not significantly associated. A moderate association of FL-Gal9, Tr-Gal9, FL-OPN, and Ud-OPN with sIL-2R was found (Figure 6A). CRP levels showed a moderate association with FL- and Ud-OPN. In CP patients, FL-Gal9 was associated with Tr-Gal9 (Figure 6B). FL-Gal9 and FL-OPN were not associated with any other inflammatory markers. However, Ud-OPN and Tr-Gal9 had a moderate association with CRP, sIL-2R, ferritin and d-dimer. Tr-Gal9 also showed a high and moderate correlation with creatinine and B2M, respectively. However, these levels were not associated with blood urea nitrogen (data not shown). Ud-OPN and Tr-Gal9 had a moderate negative correlation with SpO_2_ and S/F ratio. A weak negative association between CRP and SpO_2_ with S/F ratio and a moderate positive association between CRP levels with sIL-2R and d-dimer was found.

In the ID group, FL-Gal9 was negatively associated with FL-OPN but there was no association of the cleaved form with the FL form (Figure 6C). Ud-OPN, sIL-2R and d-dimer had a strong negative association with SpO_2_. We also observed a positive association of Ud-OPN with sIL-2R and d-dimer, which indicates that the cleavage of OPN may be associated with lung involvement, immune activation or coagulopathy in the ID group.

Notably, FL-OPN and FL-Gal9 showed negative associations, which might indicate that the responses of OPN and Gal-9 could be different in bacterial infections from viral infection. A negative association of Tr-Gal9 with CRP and a high Ud-OPN/FL-OPN ratio in ID (Appendix A) suggested this possibility.

### 2.6. Time Course of Inflammatory, Coagulation, Kidney and Respiratory Indicators during TCZ Therapy

Of 25 patients, 11 were given TCZ. The samples collected before therapy, after 4 days, 8 days, and at discharge (15–36 days) were analyzed. The values of various indicators of changes over time in each patient being treated with TCZ therapy showed the decrease of the values of FL-Gal9 (A), Tr-Gal9 (B), Ud-OPN (D), CRP (E), and the increase of the values of lymphocyte numbers (F) and S/F ratios (H) (Appendix A). In the analysis of each group, at day 4, the levels of FL-Gal9 and Tr-Gal9 decreased to 13.8% and 9.8%, respectively, but were not significant as compared with the value at before therapy. A significant reduction was observed at 15–36 days of FL-Gal9 (93.4%) and Tr-Gal9 (68.7%) (Figure 7A,B). The levels of FL-OPN did not change significantly; however, a significant reduction of Ud-OPN was observed in all the samples at day 4 (47.5%), day 8 (51.6%), and 15–36 days (65.2%) (Figure 7C,D). The marked reduction of CRP levels was also observed at day 4 (70.6%), day 8 (95.4%), and 15–36 days (99.2%), and lymphocyte numbers significantly increased (Figure 7E,F)). An apparent increase of SpO_2_ and S/F ratio during TCZ was seen but was not significant (Figure 7G,H). Levels of B2M, sIL-2R, and ferritin did not change significantly (data not shown). To confirm that these changes could be attributed with TCZ, these indicators were also monitored in the sample collected from 8 patients without TCZ therapy but treated with other drugs given in the TCZ group (Appendix A). Due to the lack of cytokine storm, these patients were discharged earlier and the data from 15–36 days were not available. Apparent reduction in the levels of FL-Gal9, Ud-OPN, and CRP was observed, but was not statistically significant.

## 3. Discussion

Here, we investigated whether the plasma levels of Gal-9 (FL- or Tr-) or OPN (FL- or Ud-) reflected the severity of disease in COVID-19-infected individuals and the efficacy of TCZ treatment.

OPN levels were significantly higher in COVID-19-infected severe patients compared with non-severe cases [45]. Gal-9 levels were elevated in subjects infected with COVID-19 [16,17] or dengue febrile illness and reflected the severity of the disease [25]. We investigated these proteins because OPN protects macrophages from apoptosis [46] and enhances Th1-mediated inflammatory responses [47], whereas Gal-9 induces apoptosis [48]. Very recently, Gal-9 induces autophagy by activating AMP-activated protein kinase (AMPK) [49]. Our results are novel because we measured the FL and cleaved forms of the products. ROC analysis demonstrated that FL-OPN and Ud-OPN had higher AUC values in the CV group compared with the HC group. In the CP group, Ud-OPN (1.00) had the highest AUC value followed by Tr-Gal9 (0.99), indicating that the cleaved forms were more specific in CP patients. Notably, FL- and Ud-OPN had the highest AUC values (1.00) in the ID group.

We previously reported that Tr-Gal9 was noticeable with AUC values of 0.9991 and 1.0000 in TB vs. AIDS and in TB vs. AIDS/TB, respectively [35]. COVID-19 develops over a shorter period compared with AIDS or AIDS/TB, and we observed the metabolism of these proteins in vivo over a very short period. It is important to detect the development of pneumonia in COVID-19 patients quickly to prevent their death. For this purpose, we studied nine markers that are often used in clinical practice and compared them with Gal-9 and OPN. ROC analysis between the CV and CP groups showed that CRP (0.94) had the highest AUC value followed by Tr-Gal9 (0.89), and ferritin (0.88). Ud-OPN had a higher AUC value (0.81) than frequently used biomarkers including sIL-2R (0.76), d-dimer (0.72), and B2M (0.67). Spearman’s analysis of the CV group showed a weak negative association of FL-OPN with the SpO_2_ and S/F ratio. The negative associations of Tr-Gal9 and Ud-OPN with the SpO_2_ and S/F ratio were more prominent in the CP group than the CV group, which indicates that the cleaved form reflects pulmonary involvement. In the CV group, the FL- and cleaved forms of OPN and Gal-9 moderately correlated with sIL-2R. It is known that these molecules might have been shed from activated T cells because Gal-9 and CD 25 are expressed by lymphocytes from COVID-19 patients [50,51]. The associations of Tr-Gal9 and Ud-OPN with sIL-2R were greater than those of FL-Gal9 and FL-OPN in in the CP group, which indicates that the cleavage of these molecules was highly active in the CP group. A previous study reported that interferon γ-induced protein 10, monocyte chemotactic protein-3, IL-1 receptor antagonist, IL-6, IL-8, IL-10, sIL-2R, IL-1^®^, IL-4 and IL-18 might be involved in the major biological processes of severe COVID-19 patients and reflect the level of systemic hyperinflammatory state [50]. Additionally, M-CSF and HGF are proposed to be involved in the major biological processes of severe COVID-19, mirroring the level of systemic hyperinflammatory state [38].

More recently, six proteins (IL-6, Cytoskeleton-associated protein 4, Gal-9, Interleukin-1 receptor antagonist, Leukocyte Immunoglobulin-Like Receptor B4, and Programmed cell Death ligand 1) among 368 proteins, were identified to be associated with disease severity [51].

Therefore, Tr-Gal9 and/or Ud-OPN might be used as biomarkers of disease severity in the CP group. Notably, Tr-Gal9 levels had a negative association with lymphocytes. Lymphopenia is a common feature of severe COVID-19 and is thought to be caused, at least in part, by massive lymphocyte death mediated by pyroptosis [52]. This would release Gal-9, a cytoplasmic protein, into the extracellular space, acting as a danger-associated molecular pattern to transduce danger signals to activate innate and adaptive immune systems. T cells in active COVID-19 disease were reported to be inactive with an exhausted phenotype characterized by the expressions of PD-1^+^ and Tim-3^+^ [50]. Recently, Gal-9 was shown to bind to PD-1 as well as Tim-3 to form a lattice structure comprised of the three proteins, which was necessary for T cells to maintain the exhausted phenotype [53]. Therefore, elevated levels of Tr-Gal9 might reflect the severity and exhaustion of T cells. 

Furthermore, the negative correlation of Tr-Gal9 with SpO_2_ suggests the involvement of Gal-9 in pneumonia. Ultra-high resolution CT analysis showing local lung volume loss caused by alveolar collapse is a hallmark of COVID-19 pneumonia [54]. Alveolar collapse is caused by alveolar cell damage, in which alveolar cells in the lungs undergo programmed cell death [55]. In addition, cytokine storm is involved in apoptosis of the alveolar cells [56]. Therefore, whether Gal-9 can induce apoptosis of the alveolar epithelial cells should be determined in the future.

Ud-OPN but not FL-OPN had a strong negative correlation with SpO_2_ and the involvement of monocytes in the production of OPN in COVID-19 pneumonia was previously proposed [18]. Furthermore, OPN but not CRP correlated with the severity of community-acquired pneumonia [57]. OPN knockout mice survived longer than wild-type mice upon intranasal infection of *Streptococcus pneumoniae*, which indicates the detrimental effect of OPN on anti-bacterial defense in the pulmonary compartment during pneumonia [58]. During COVID-19 infection, OPN-induced furin may enhance the entry of SARS-CoV2 to target cells [59]. Furthermore, OPN levels were associated with the severity of COVID-19 infection and white blood cell numbers [45]. Recently, activated neutrophil markers including IL-8 and MMP-8 were proposed to distinguish patients at risk of future clinical decompensation, although OPN and Gal-9 were not examined [60]. The associations of IL-8 and OPN were previously reported in patients with tuberculosis [61]. 

These findings suggest that Gal-9 and OPN might be therapeutic targets to ameliorate COVID-19 infection. A new brefelamide derivative inhibits the synthesis of OPN and Gal-9, which might serve as a therapeutic agent for COVID-19 [62].

The association of Tr-Gal9 with creatinine and d-dimer suggests their involvement in kidney diseases and coagulopathy, which are often associated with COVID-19 infection [33,63]. None of our patients met the criteria for acute kidney injury or severe coagulopathy, and Tr-Gal9 levels did not correlate with neutrophil numbers, blood urea nitrogen, or prothrombin time (data not shown); however, Gal-9 was reported to be involved in kidney diseases [64] and atherosclerotic stroke [65]. Therefore, further analysis of the role of Gal-9 in COVID-19 coagulopathy is necessary.

Our results suggest that the moderate association of FL-OPN with CRP and sIL-2R, observed in the CV group was decreased in the CP group, which indicates that it might be rapidly cleaved in inflammatory environments. OPN was released in the form of exosome from a lipopolysaccharide-stimulated macrophage cell line and enhanced encapsulation of FL-OPN was reported, suggesting that the exosomes may be a suitable vehicle for transferring OPN to target cells [66]. We should determine whether OPN needs to be present in plasma in the form of exosomes to exert its biological functions. 

Multiple humoral factors have been claimed to be associated with the severity or morbidity in COVID-19 patients. To assess their exact roles, the application of the strategy used in the VA COVID-19 (VACO) should be developed using nationwide medical administrative data [67]. In HIV study, higher Veterans Aging Cohort Study index scores were associated with higher levels of neopterin, cystatin C, tumor necrosis factor receptor 1 and Gal-9 in individuals under therapy [68]. 

A major finding of this study was the cleavage of OPN and Gal-9 in CP patients. The cleavage of OPN and Gal-9 by various proteases including MMPs and thrombin and the involvement of these proteases in inflammatory responses was previously reported. Among various inflammatory markers, MMP-9 was strongly associated with the P/F ratio and distinguished between patients with and without respiratory failure [69]. In addition, an increase in MMP-9 reflects neutrophil activation and may be associated with the development of thrombotic events [70]. It is not clear whether the biological effects of OPN and Gal-9 might be inactivated by their cleavage, because the cleaved products also have distinct biological activities [23]. These complex interactions of biological molecules may be important for the resolution of inflammation. 

The patients studied here were treated with multiple agents including favipiravir (FAV) [71]. It was reported that TCZ with or without FAV effectively improved pulmonary inflammation and symptoms of COVID-19 patients [72,73]. A recent randomized trial of hospitalized patients with severe COVID-19 pneumonia reported the potential benefit of TCZ in the period until hospital discharge and during ICU stay [13]. Evaluation of a minority patients treated with TCZ showed that TCZ lowered the composite rate of mechanical ventilation or death [74]. The optimal time to prescribe TCZ was reported to be the early stages of inflammation and the initial reduction in O_2_ saturation [75]. All patients survived in this study, probably due to early administration of TCZ and other possibly effective agents such as azithromycin [76], ciclesonide [77], nafamostat mesilate [78], and FAV. It is better to follow the new guideline of TCZ therapy in the future [79]. TCZ binds to the IL-6 receptor to inhibit the IL-6 signaling pathway; however, IL-6 levels were increased after treatment with TCZ [10]. This study also confirmed a decrease in Tr-Gal9 and Ud-OPN levels after TCZ therapy, although sIL-2R, ferritin, and B2M levels were not significantly reduced. The reduction seen in TCZ-treated patients was not observed in patients not treated with TCZ, but more patients are needed to conclude the results. CRP levels declined so rapidly that they were not suitable for monitoring TCZ therapeutic effects, because CRP synthesis is partially dependent on IL-6 [37]. The prominent decrease in Ud-OPN cannot be explained by the reduced protein expression, because FL-OPN levels did not decrease. It is possible that IL-6-dependent stat3-mediated protease activation [80] is involved in the generation of Ud-OPN and was impaired by TCZ therapy. This is a novel finding demonstrating cleavage of OPN by IL-6-dependent proteases in vivo. Therefore, Tr-Gal9 might be suitable to monitor the therapeutic effect of TCZ on COVID-19 patients with cytokine storm. However, we emphasize that other monitoring tools should also be used because TCZ increased d-dimer levels [72] and the development of multiorgan failure was reported [81]. Alternatively, Ud-OPN could be produced by intrinsic protease activities in COVID-19 pneumonia patients. FL-OPN was not reduced by TCZ therapy probably because not only IL-6-dependent pathway [82], but also IL-6 independent of OPN synthesis was involved [83]. It should be noted that persistent elevation of OPN was also observed after ATR therapy in AIDS patients [84], and chemotherapeutic drugs may induce the synthesis of OPN [85].

In summary, we showed that OPN and Gal-9 were released in COVID-19-infected patients, and their cleaved products might be useful biomarkers for assessing the severity of COVID-19 pneumonia. Furthermore, levels of the cleaved products might be useful to monitor the therapeutic effect of TCZ in cytokine storm complicated in CP patients.

The disadvantages of this study included the relatively small number of patients enrolled from a single medical hospital. None of the treated patients received TCZ alone because they received multiple medications.

## 4. Materials and Methods

### 4.1. Study Design and Participants

This was a cross-sectional analytical study, and patient samples were collected at SCH, Sendai, Japan, from July 2020 to October 2020. Among 4897 patients in the out-patient department, 157 febrile patients were screened by a SARS-CoV-2 PCR test using a sample obtained from a nasopharyngeal swab as previously described [72]. Among 49 positive cases, one patient who had been suffering from pulmonary embolism was excluded. Among 108 negative cases, informed consent was obtained from 18 cases, and 14 patients who were clinically suspected of bacterial infection were enrolled (Figure 1). Of the 23 CV patients, 12 were hospitalized and the remaining 11 were outpatients. 

Various drugs were given to the patients, including azithromycin (500 mg/day), ciclesonide (200 µg inhaler; two inhalations per day) [77], nafamostat mesilate (0.06 mg/kg/day) [78], FAV (3600 mg on the first day, 1600 mg thereafter) [71] and TCZ (8 mg/kg) administered by a single intravenous injection when patients showed signs of cytokine storm [9,10].

Four and thirteen patients in the CV and CP groups, respectively, did not show signs of cytokine storm and were treated with the above drugs without TCZ. Other patients in the CV group were not treated because of a lack of disease symptoms. Twelve patients suspected of cytokine storm in the CP group were treated with five or six drugs including TCZ. Plasma from peripheral blood samples was obtained in EDTA tubes at admission, during therapy, and before discharge. Patients in the ID group were examined only at admission. All patients had improved symptoms and were discharged except for two patients in the ID group, who died during hospitalization. EDTA plasma was stored at −80 °C to measure FL-Gal9, Tr-Gal9, FL-OPN, and Ud-OPN. Thirty normal human plasma samples that were negative for HIV, SARS-CoV-2, and hepatitis B and C viruses were obtained as HC from Bioivt (Hicksville, NY, USA). Their age ranged from 19–64 years with an average of 43.7. There were 23 women accounting for 76%.

### 4.2. Laboratory Analyses

The SpO_2_, SpO_2_/FiO_2_ ratio, conventional laboratory analyses, and chest CT were obtained through Sendai City Hospital (Table 1) as described [72].

### 4.3. Determination of FL-OPN and Ud-OPN

To identify FL-OPN, an ELISA kit (JP27158, IBL, Gunma, Japan) was used. Ud-OPN was determined using the Human OPN DuoSet ELISA Development System kit (DY1433, R&D Systems, Minneapolis, MN, USA) [27].

### 4.4. Determination of FL-Gal9 and Tr-Gal9

FL-Gal9 was measured using a human Gal-9 ELISA kit (GalPharma Co., Ltd., Takamatsu, Japan). An ELISA for Tr-Gal9 was constructed using two monoclonal antibodies against the N-terminal carbohydrate-recognition domain of human Gal-9, in which 9S2-3 (GalPharma) and biotinylated ECA8 (MBL, Nagoya, Japan) were used as the capture and the detection antibodies, respectively, as described previously [34,35]. Ten mM lactose in dilution buffers for specimen and detection antibodies were used to prevent Gal-9 from forming a complex with carbohydrates.

### 4.5. Ethical Statements

This study adhered to the ethical considerations and ethical principles set out in relevant guidelines, including the Declaration of Helsinki, WHO guidelines, International Conference on Harmonization-Good Clinical Practice, Data Privacy Act of 2012, and National Ethics Guidelines for Health Research 2017. This research was approved by the Ethics Review Unit of Sendai City Hospital (SCH 338-20202001). Written informed consent was obtained from all patients prior to enrollment. 

### 4.6. Statistical Analysis

Statistical analysis was performed using R Statistical Software (version 3.5.3; R Foundation for Statistical Computing, Vienna, Austria) and Prism 8 (GraphPad software, San Diego, CA, USA). The Mann–Whitney U-test and Kruskal–Wallis test were used to assess the differences between two groups and among multiple groups, respectively. Correlations between a data set were examined using Spearman’s rank correlation coefficient. ROC analysis, including the corresponding AUC calculation, was conducted to analyze the ability of biomarkers to discriminate between a selected pair of the HC, CV, CP, and ID groups. 

## 5. Conclusions

FL-OPN, Ud-OPN, FL-Gal9, Tr-Gal9, and commonly used inflammatory and respiratory markers of patients with CV, CP, and ID were analyzed. FL-OPN, Ud-OPN, and Tr-Gal9 levels in the CV group were significantly higher than in the HC group. ROC analysis showed that the cleaved form had a high discriminating power between the HC and CV or CP groups and between the CV and CP groups. Spearman’s analysis showed that FL-OPN and Ud-OPN had moderate associations with CRP, sIL-2R, and d-dimer but FL-Gal9 and Tr-Gal9 showed the moderate association only with sIL-2R in the CV group. However, Tr-Gal9 and Ud-OPN levels were more associated with inflammatory or respiratory functional parameters compared with FL-Gal9 or FL-OPN in the CP group. The levels of FL-Gal9, Tr-Gal9, Ud-OPN, and CRP were significantly decreased in the TCZ-treated group. The decrease of Ud-OPN could be ascribed to the impaired synthesis of IL-6-dependent protease, because the FL-Gal9 levels did not decrease. Therefore, the cleaved forms of Gal-9 and OPN are useful to assess the severity of COVID-19 pneumonia and Tr-Gal9 may be useful to determine the therapeutic effect of TCZ in COVID-19 pneumonia. Further studies are necessary to support our hypothesis.

## Figures and Tables

**Figure 1 ijms-22-04978-f001:**
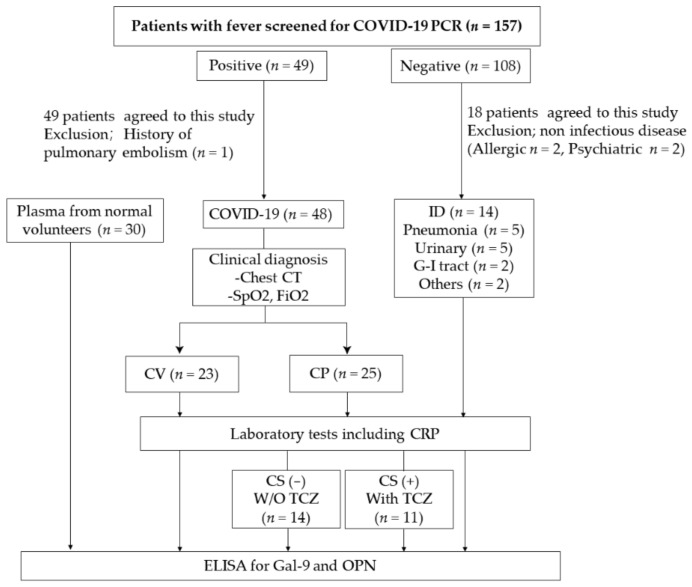
Patients over the age of 20 years participated in this study. Adults who did not have sufficient judgment, were unconscious, or who need consideration for the name of the disease were excluded. Laboratory tests included C-reactive protein (CRP), chest CT and SpO_2_. COVID-19-infected patients with mild clinical symptoms (CV), COVID-19 patients associated with pneumonia (CP), and patients with bacterial infection (ID) were studied. Patients suspected of having cytokine storm (CS) were treated with TCZ. The treatment policy depended on the patient’s clinical situation.

**Figure 2 ijms-22-04978-f002:**
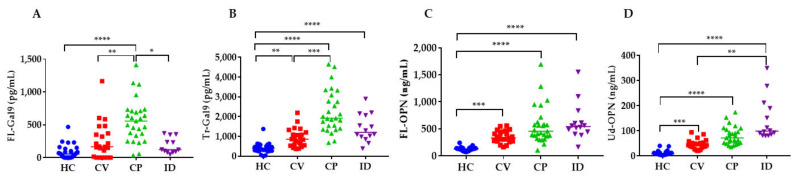
Levels of FL-Gal9 (**A**), Tr-Gal9 (**B**), FL-OPN (**C**), and Ud-OPN (**D**) in CV, CP, ID, and HC. HC: healthy control; CV: COVID-19 infection with mild clinical symptoms; CP: COVID-19 associated with pneumonia; ID: infectious diseases FL-Gal9; Full-length Gal-9, Tr-Gal9; truncated Gal-9, FL-OPN; full-length OPN, Ud-OPN; undefined OPN, **** *p* < 0.0001, *** *p* < 0.001, ** *p* < 0.01, * *p* < 0.05.

**Figure 3 ijms-22-04978-f003:**
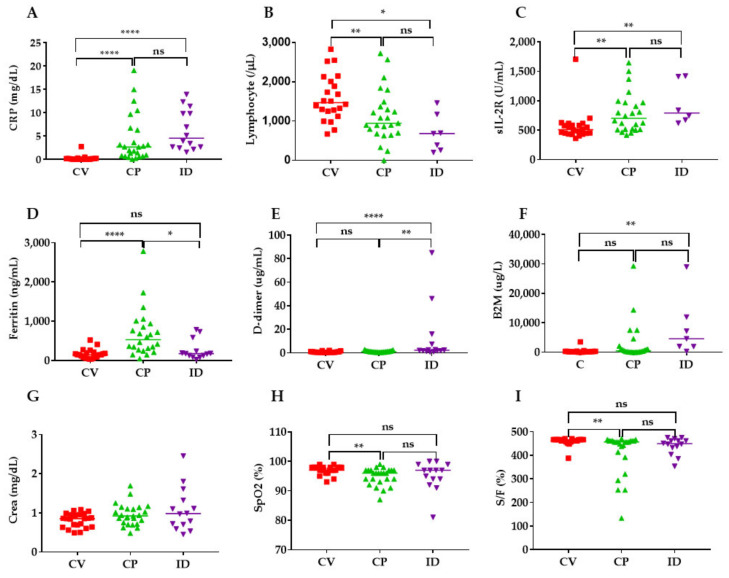
Levels of inflammatory, coagulation, kidney and respiratory indicators in COVID-19 patients (CV, CP) and patients with bacterial infection (ID). Only those with data are shown in the figure. CRP (**A**), Lymphocyte number (**B**), sIL-2R; soluble IL-2 receptor α (**C**), Ferritin (**D**), D-dimer; d-dimer (**E**), B2M; urinary β_2_-microglobulin (**F**), Crea; creatinine (**G**), SpO_2_; peripheral capillary oxygen saturation (**H**), S/F; SpO_2_/FiO_2_ ratio (**I**). CRP, ferritin, and creatinine were measured in plasma and d-dimer was measured in serum. **** *p* < 0.0001, ** *p* < 0.01, * *p* < 0.05; ns; not significant.

**Figure 4 ijms-22-04978-f004:**
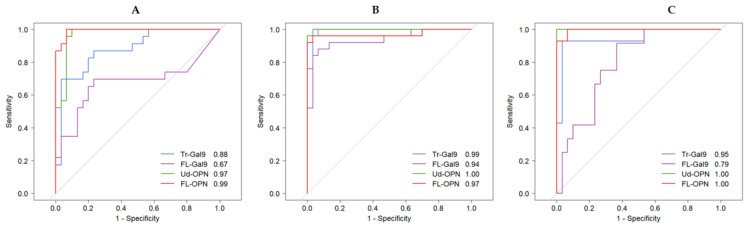
ROC analysis of Gal-9 (FL-Gal9 and Tr-Gal9) and OPN (FL-OPN and Ud-OPN) between the HC and CV groups (**A**), the CP group (**B**) and the ID group (**C**).

**Figure 5 ijms-22-04978-f005:**
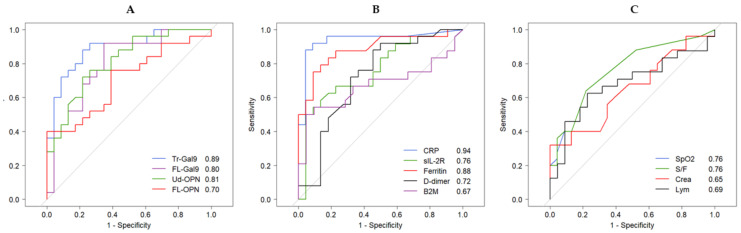
ROC analysis of inflammatory, coagulation, kidney and respiratory indicators between the CV and CP groups. Gal-9 and OPN (**A**), inflammatory markers (**B**), and respiratory, kidney and hematological markers (**C**).

**Figure 6 ijms-22-04978-f006:**
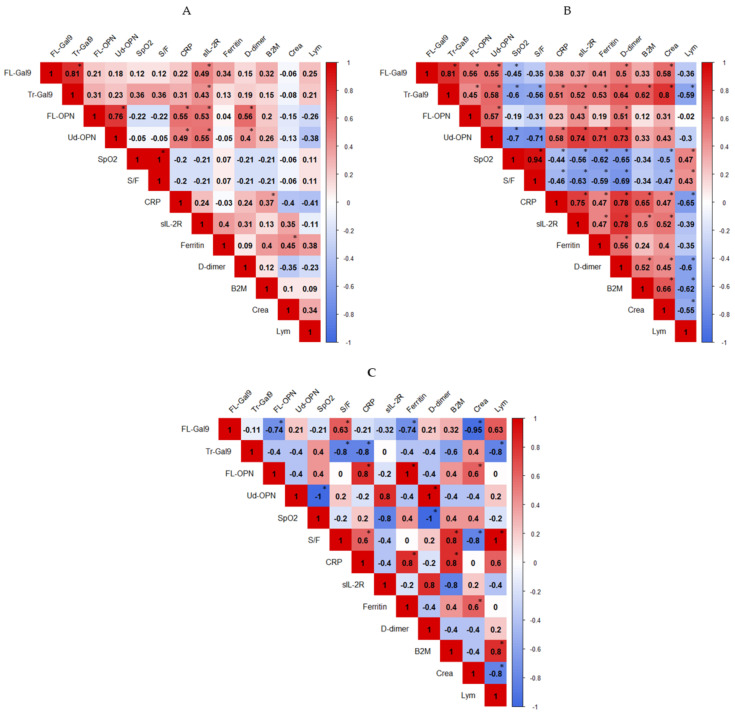
Associations of studied indicators in the CV (**A**), CP (**B**), and ID (**C**) groups. The correlation was measured by the Spearman *t*-test. The correlation R-value is written in each well and displayed as colors ranging from blue to red as shown in the legend key. *p*-value is written significant as * *p* < 0.05.

**Figure 7 ijms-22-04978-f007:**
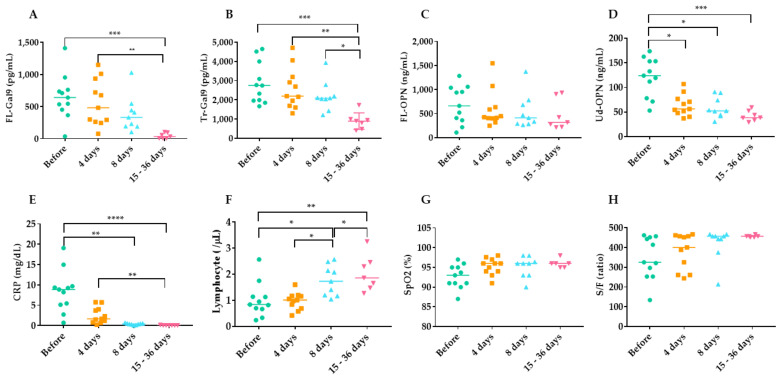
Effects of TCZ therapy on markers. Only those with data are shown in the figure. FL--Gal9 (**A**), Tr-Gal9 (**B**), FL-OPN (**C**), Ud-OPN (**D**), CRP (**E**), Lymphocyte (**F**), SpO2 (**G**), S/F ratio (**H**). TCZ; tocilizumab, **** *p* < 0.0001, *** *p* < 0.001, ** *p* < 0.01, * *p* < 0.05.

**Table 1 ijms-22-04978-t001:** Patient demographics.

		Reference Range	CV (*n* = 23)	CP (*n* = 25)	ID (*n* = 14)	*p* Value
Basic information	Age (range)		36.7 (19–102)	54.8 (20–99)	70.11 (23–90)	0.0002
	Male		13 (56.5%)	22 (88%)	7 (50%)	<0.0001
Blood routine test	WBC ^1^ (10^3^/µL)	3.7–8.5	4.82 (1.44) ^2^	5.2 (1.22)	9.08 (4.71)	0.0046
	PLT ^3^ (10^4^/µL)	0.15–3.55	22.7 (4.1)	20.6 (7.42)	20.9 (5.11)	0.1894
	RBC ^4^ (10^6^/µL)	3.9–5.3	4.93 (0.80)	4.72 (0.50)	4.05 (1.06)	0.0137
Biochemical test	ALT ^5^ (U/L)	3–40	21 (16.6)	63.6 (55.1)	21.6 (13.3)	<0.0001
	AST ^6^ (U/L)	8–35	21.6 (6.88)	51.6 (32.4)	31.9 (24.7)	<0.0001
	CRP ^7^ (mg/dl)	0.00–0.3	0.24 (0.56)	4.36 (5.12)	6.31 (4.31)	<0.0001
	Alb ^8^ (g/dl)	3.8–5.2	4.54 (0.60)	3.90 (0.56)	3.36 (0.76)	<0.0001
Coagulation system	PT ^9^ (sec)	11.2	12.0(0.85)	11.7 (1.14)	11.6 (6.34)	0.0893

^1^ WBC: white blood cell count; ^2^ (): the numbers in parentheses are the standard deviation, ^3^ PLT: platelet count; ^4^ RBC: red blood cell count; ^5^ ALT: alanine aminotransferase; ^6^ AST: aspartate aminotransferase; ^7^ CRP: C-reactive protein; ^8^ Alb: albumin; ^9^ PT: prothrombin time.

**Table 2 ijms-22-04978-t002:** Clinical characteristics of patients in this study.

		CV (*n* = 23)	CP (*n* = 25)	ID (*n* = 14)
Complications	High blood pressure	0	9	2
	Hyperlipidemia	0	2	2
	Diabetes mellitus	1	7	1
	Coronary artery disease	0	0	1
	Cerebral infarction	0	1	2
Clinical symptoms	Cough	6	18	3
	Diarrhea	4	10	1
	Dyspnea	0	8	4
	Fever	13	17	17
Clinical classification	Asymptomatic	4	0	0
	Mild	16	2	1
	Moderate	1	3	6
	Severe	2	16	7
	Critical	0	4	0

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
