# Peer review of "High Levels of the Cleaved Form of Galectin-9 and Osteopontin in the Plasma Are Associated with Inflammatory Markers That Reflect the Severity of COVID-19 Pneumonia"

_ijms, 2021, doi:10.3390/ijms22094978_

Round 1

Reviewer 1 Report

High levels of the cleaved form of galectin-9 and osteopontin in the plasma are associated with inflammatory markers that reflect the severity of COVID-19 pneumonia and decline after the tocilizumab therapy by Bai et al.

This is a timely and interesting study considering the COVID-19 pandemic. Authors have shown that Gal-9 and osteopontin are elevated in the plasma of COVID-19 patients. They have also provided some correlational analysis for their contribution to the disease severity. Although it’s an important study, lacks a mechanistic depth and its very superficial.

Major concerns:

  1. The authors have measured the plasma levels of Gal-9 eight the FL or Truncated, and OPN in 11 patients who took TCZ serially for 15-36 days, and they showed that FL-Gal9, Tr-Gal9, and Ud-OPN concentrations significantly decreased overtime. They attributed the decline in these markers to TCZ therapy and even they reflected this conclusion in the title. Firstly, the main concern is that those patients did not get the TCZ therapy as the only medication. It is mentioned in the study that they got the Corticosteroid and other medications at the same time. The attribution of the Gal9 and OPN decrement to just TCZ therapy doesn’t seem justified as the effect of other medications have not been elucidated. Secondly, this study lacks a control group for TCZ treatment. There is a need for a control group that didn’t get the CS and TCZ. If they measured these markers, did they see any decline over time in the absence of TCZ? We believe Gal-9 plasma levels and other measured pro-inflammatory factors decline over time in the absence of any treatment. A control group not getting TCZ therapy is necessary to compare the Gal9 and OPN levels over time. Since such control group is missing the direct effect of TCZ on decreasing plasma Gal9 and OPN levels can be misleading. Especially it has been included in the title without properly being investigated.
  2. Why the number of ID patients are changing depending on the assay? Why some are excluded in the analysis (e.g. CRP n=14 but IL-2R, B2m and lymphocytes n=6).
  3. How authors justify the inclusion of non-respiratory associated conditions with COVID-19 infection (e.g. urinary and the GI, others)?

Other comments:

The introduction lacks some basic info regarding the analyzed markers such as CRP, Ferritin, sIL-2R, D-dimer, B2M, Creatinine as inflammatory, coagulopathy/cardiac, or kidney function markers, and their clinical significance.

  1. 1: Are normal volunteers were age-and-sex matched?
  2. The mild cases (CV) are outpatients or hospitalized or combination of both?
  3. 2A: FL-Gal9 unit on Y axis is ng/ml or pg/ml? In Fig.7A on Y axis it is different and expressed as pg/ml?
  4. 2: A graph comparing the ratio of Tr-Gal9/FL-Gal9 and Ud-OPN/FL-OPN really helps to understand the ratios of these markers changing between groups.
  5. Page 5, Paragraph 1, Line 141: Delete “in the CV group” as it is repeating in line 142.
  6. 3: Use the same color coding as used for Fig.1.
  7. 3: Label each graph with letters (A, B, C, D,..)
  8. 3: In the legend, describe the source of CRP, Ferritin, Creatinine, D-dimer as plasma?
  9. 3: Place B2M and Creatinine graphs next to each other as they are representative of kidney function.
  10. Page7, Paragraph 2, Lines 186-188: Are d-dimer and sIL-2R indicators of lung involvement? In ID group there were different patients with different sources of infection as labeled in Fig.1. There were only 5 patients with pneumonia. How do you relate theses markers to the lung involvement, as discussed above?
  11. Page 7, Paragraph 3, line189-190: The justification is confusing, it requires clarification.
  12. 7, Legend: Markers in (B) reported as inflammatory markers but only CRP is inflammatory marker and others (Lymphocyte, SpO2 and S/F) are not. It is suggested to label each graph with a different letter and explain in the legend separately.
  13. 7: It is suggested to use other colors for these graphs as they show time points and not different groups of patients. Using the same color coding for patient different groups is confusing.
  14. 7: It will be more informative to show the changes for each individual patient change using connected lines over time.
  15. 7: The number of patients as dots are not consistent during time points (Decreasing?). Explain about it.
  16. Page11, Paragraph 2, Line 340: Gal9 were synthesized?? Can be replaced by released.

In general the quality/resolution of figures should be improved.

Author Response

High levels of the cleaved form of galectin-9 and osteopontin in the plasma are associated with inflammatory markers that reflect the severity of COVID-19 pneumonia and decline after the tocilizumab therapy by Bai et al.

This is a timely and interesting study considering the COVID-19 pandemic. Authors have shown that Gal-9 and osteopontin are elevated in the plasma of COVID-19 patients. They have also provided some correlational analysis for their contribution to the disease severity. Although it’s an important study, lacks a mechanistic depth and its very superficial.

Major concerns:

  1. The authors have measured the plasma levels of Gal-9 eight the FL or Truncated, and OPN in 11 patients who took TCZ serially for 15-36 days, and they showed that FL-Gal9, Tr-Gal9, and Ud-OPN concentrations significantly decreased overtime. They attributed the decline in these markers to TCZ therapy and even they reflected this conclusion in the title. Firstly, the main concern is that those patients did not get the TCZ therapy as the only medication. It is mentioned in the study that they got the Corticosteroid and other medications at the same time. The attribution of the Gal9 and OPN decrement to just TCZ therapy doesn’t seem justified as the effect of other medications have not been elucidated. Secondly, this study lacks a control group for TCZ treatment. There is a need for a control group that didn’t get the CS and TCZ. If they measured these markers, did they see any decline over time in the absence of TCZ? We believe Gal-9 plasma levels and other measured pro-inflammatory factors decline over time in the absence of any treatment. A control group not getting TCZ therapy is necessary to compare the Gal9 and OPN levels over time. Since such control group is missing the direct effect of TCZ on decreasing plasma Gal9 and OPN levels can be misleading. Especially it has been included in the title without properly being investigated. 

R Thank you very much for the important questions. We analyzed the changes of indicators of 8 patients without TCZ therapy. Because these patients did not suffer from cytokine storm, we could follow only 8 days but the significant changes such as the reduction of the levels of Ud-OPN and CRP observed in patients with TCZ therapy were not observed (Supplementary Figure S3).   Although we need more numbers of patients to conclude these results. (lines 248-253)

  1. Why the number of ID patients are changing depending on the assay? Why some are excluded in the analysis (e.g. CRP n=14 but IL-2R, B2m and lymphocytes n=6).

R. Thank you for your comments.  Only those with data are shown in the figure. Accordingly, legends of Fig 3 were changed.

3 How authors justify the inclusion of non-respiratory associated conditions with COVID-19 infection (e.g. urinary and the GI, others)?

R  Thank you for your question. ID group are patients with bacterial infection and negative for COVID-19 PCR (Fig.1). Since the number of pneumonia patients was insufficient, those infected with other organs were also included.

Other comments:

The introduction lacks some basic info regarding the analyzed markers such as CRP, Ferritin, sIL-2R, D-dimer, B2M, Creatinine as inflammatory, coagulopathy/cardiac, or kidney function markers, and their clinical significance. 

R Thank you for your constructive comments we added the basic information of above indicators and stated its significance (Lines106-123)

  1. 1: Are normal volunteers were age-and-sex matched? 

R Thank you for inquiry. Their age and sexuality were described (Lines 430-1)

  1. The mild cases (CV) are outpatients or hospitalized or combination of both?

R  Thank you for your inquiry.

Of the 23 CV patients, 12 were hospitalized and the remaining 11 were outpatients (Line 414).

  1. 2A: FL-Gal9 unit on Y axis is ng/ml or pg/ml? In Fig.7A on Y axis it is different and expressed as pg/ml?

R Thank you for your comments. It should be pg/ml and it was corrected. (Fig 2A)

  1. 2: A graph comparing the ratio of Tr-Gal9/FL-Gal9 and Ud-OPN/FL-OPN really helps to understand the ratios of these markers changing between groups.

R Thank you for an important comment. The ratio was analyzed but we could not see significant difference between CV and CP (Lines 159-63), therefore the figure was provided as supplementary figure 1.

  1. Page 5, Paragraph 1, Line 141: Delete “in the CV group” as it is repeating in line 142.

R Thank you for your correction. It was deleted.

  1. 3: Use the same color coding as used for Fig.1.

R  Thank you for your suggestion.  The color of Fig 3 was changed.

  1. 3: Label each graph with letters (A, B, C, D,..)

R: It was changed. Accordingly, figure legend was changed.

  1. 3: In the legend, describe the source of CRP, Ferritin, Creatinine, D-dimer as plasma?

R  Thank you for your comment. It was described as CRP, ferritin and creatinine were measured in plasma and d-dimer was measured in serum (lines 181-2).

  1. 3: Place B2M and Creatinine graphs next to each other as they are representative of kidney function.

Thank you for your comments. The order of the items in the figure was changed according to each purpose.

  1. Page7, Paragraph 2, Lines 186-188: Are d-dimer and sIL-2R indicators of lung involvement? In ID group there were different patients with different sources of infection as labeled in Fig.1. There were only 5 patients with pneumonia. How do you relate theses markers to the lung involvement, as discussed above?

Thank you for your valuable comments. It is true that ID group has various infectious disease but strong negative association of Ud-OPN, d-dimer and sIL-2R suggested that these indicators may reflect lung involvement and patients with other organs often show lung impairment but other reasons also exist. Therefore, it was changed as “the cleavage of OPN may be associated with lung involvement, immune activation or coaglulopathy in the ID group.(Lines 220-2)

  1. Page 7, Paragraph 3, line189-190: The justification is confusing, it requires clarification.

R: Sorry about confusing description. It was changed as Notably, FL-OPN and FL-Gal9 showed negative associations which might indicate the responses of OPN and Gal-9 could be different in bacterial infections from viral in-fection.  A negative association of Tr-Gal-9 with CRP and a high Ud-OPN/FL-OPN ratio in ID  (Supplementary Figure S1 B) suggested this possibility.(Lines 223-6)

.

  1. 7, Legend: Markers in (B) reported as inflammatory markers but only CRP is inflammatory marker and others (Lymphocyte, SpO2 and S/F) are not. It is suggested to label each graph with a different letter and explain in the legend separately.

R:Thank you for your comments. The labels were changed as instructed.

  1. 7: It is suggested to use other colors for these graphs as they show time points and not different groups of patients. Using the same color coding for patient different groups is confusing.

R Thank you for your comments. The colors were changed as instructed

  1. 7: It will be more informative to show the changes for each individual patient change using connected lines over time.

R: Thank you very important comment. The figure of each patients was also provided as supplementary figure 2 and was described (Lines 235-8).

  1. 7: The number of patients as dots are not consistent during time points (Decreasing?). Explain about it.

R: Thank you for your comments.  Only those with data are shown in the figure as described in the legends.

  1. Page11, Paragraph 2, Line 340: Gal9 were synthesized?? Can be replaced by released.

R: Thank you for your comments. It was changed to release (line 398).

In general the quality/resolution of figures should be improved.

R: Thank you for your comment. We tried to increase the resolution of the figures as much as possible.

Reviewer 2 Report

The manuscript submitted to IJMS entitled “High levels of the cleaved form of galectin-9 and osteopontin in the plasma are associated with inflammatory markers that reflect the severity of COVID-19 pneumonia and decline after the tocilizumab therapy” is an original article which aim to evaluate galectin-9 (Gal9) and osteopontin (OPN) as inflammatory biomarkers to monitor the severity of pathological inflammation and the therapeutic effects of tocilizumab in COVID-19 patients with pneumonia.

On my opinion the article is interesting, well written, with good English. 

I highlighted some minor issues:     

  • minor spell check required    
  • summary of abbreviations required    
  • Introduction: This section could be improved with an update on complication related with tocilizumab administration (DOI: 10.1016/j.oraloncology.2020.104659).    
  • Results: This section has been properly prepared.    
  • Discussion: Are there other studies regarding the evaluation of other biomarkers for monitoring COVID-19 patients?Please discuss
  • Materials and Methods: This section has been properly prepared.    
  • Conclusions: Further studies are necessary to support Authors’ hypothesis.

After making the indicated changes, the article may be suitable for publication.

Author Response

Reviewer 2

The manuscript submitted to IJMS entitled “High levels of the cleaved form of galectin-9 and osteopontin in the plasma are associated with inflammatory markers that reflect the severity of COVID-19 pneumonia and decline after the tocilizumab therapy” is an original article which aim to evaluate galectin-9 (Gal9) and osteopontin (OPN) as inflammatory biomarkers to monitor the severity of pathological inflammation and the therapeutic effects of tocilizumab in COVID-19 patients with pneumonia.

On my opinion the article is interesting, well written, with good English. 

I highlighted some minor issues:     

  • minor spell check required    
  • summary of abbreviations required    
  • Introduction: This section could be improved with an update on complication related with tocilizumab administration (DOI: 10.1016/j.oraloncology.2020.104659).  

R: Two references (14,15) were added regarding the adverse effect of TCZ in  Introduction (Lines 78-80)  

  • Results: This section has been properly prepared.    
  • Discussion: Are there other studies regarding the evaluation of other biomarkers for monitoring COVID-19 patients? Please discuss
  1. We added more indicators described in Ref 50. We also suggested to use VACO analysis to identify the most important molecules, which represent the disease severity (Ref 68) and also introduced VACS analysis (Ref 69) .
  • Materials and Methods: This section has been properly prepared.    
  • Conclusions: Further studies are necessary to support Authors’ hypothesis.

R: Thank you for your comments. It was added to the conclusion (Lines 477-8).

After making the indicated changes, the article may be suitable for publication.

Round 2

Reviewer 1 Report

The authors have addressed most of my comments and improved the quality of figures. However, I am not fully convinced with the last part of the title "and decline after the tocilizumab therapy"

Considering a very small number of patients this is a strong conclusion and I am hesitant to support this conclusion. The authors should allow the readers to make a conclusion based on the content.   

Author Response

Dear Sir;

Thank you very much for your very important and careful  comment on the title. The title was changed to 

High levels of the cleaved form of galectin-9 and osteopontin in the plasma are associated with inflammatory markers that reflect the severity of COVID-19 pneumonia and apparent change in their levels after the tocilizumab therapy.

If it is not satisfactory in your mind, you can delete the last part 

"and apparent change in their levels after the tocilizumab therapy".

I leave it to your reasonable decision.

We also corrected spelling.

Looking forward to your kind reply.

Toshio
